# Contactless Fall Detection by Means of Multiple Bioradars and Transfer Learning

**DOI:** 10.3390/s22166285

**Published:** 2022-08-21

**Authors:** Vera Lobanova, Valeriy Slizov, Lesya Anishchenko

**Affiliations:** Remote Sensing Laboratory, Bauman Moscow State Technical University, 105005 Moscow, Russia

**Keywords:** bioradiolocation, remote sensing, fall detection, deep learning, transfer learning, wavelet analysis

## Abstract

Fall detection in humans is critical in the prevention of life-threatening conditions. This is especially important for elderly people who are living alone. Therefore, automatic fall detection is one of the most relevant problems in geriatrics. Bioradiolocation-based methods have already shown their efficiency in contactless fall detection. However, there is still a wide range of areas to improve the precision of fall recognition based on view-independent concepts. In particular, in this paper, we propose an approach based on a more complex multi-channel system (three or four bioradars) in combination with the wavelet transform and transfer learning. In the experiments, we have used several radar configurations for recording different movement types. Then, for the binary classification task, a pre-trained convolutional neural network AlexNet has been fine-tuned using scalograms. The proposed systems have shown a noticeable improvement in the fall recognition performance in comparison with the previously used two-bioradar system. The accuracy and Cohen’s kappa of the two-bioradar system are 0.92 and 0.86 respectively, whereas the accuracy and Cohen’s kappa of the four-bioradar system are 0.99 and 0.99 respectively. The three-bioradar system’s performance turned out to be in between two of the aforementioned systems and its calculated accuracy and Cohen’s kappa are 0.98 and 0.97 respectively. These results may be potentially used in the design of a contactless multi-bioradar fall detection system.

## 1. Introduction

In 2018, for the first time in history, persons aged 65 or above outnumbered children under five years of age globally [1]. By 2050, one in six people in the world will be over age 65 (16%), up from one in eleven in 2019 (9%), whereas one in four persons living in Europe and Northern America could be aged 65 or over [1]. The number of persons aged 80 years or over is projected to triple, from 143 million in 2019 to 426 million in 2050 [1]. Therefore, a higher awareness of the needs of elderly people should be awakened.

Due to the natural aging process, comorbidity, and medication use, different kinds of negative health changes often occur. For instance, difficulties with coordination with a higher probability of fainting and bone fragility pose a serious risk of falling with severe consequences. Meanwhile, falls are one of the main causes of disability along with sensory impairments, chronic obstructive pulmonary disease, depressive disorders, diabetes, dementia, and osteoarthritis [2]. Moreover, older people have the highest risk of death arising from a fall, and the risk increases with age [3].

Therefore, the problem of automatic fall detection has led to the wide adoption of state-of-the-art technologies for personal health monitoring.

Nowadays, there are three groups of fall detection methods [4]. The first one includes wearable sensor-based methods which use various contact sensors to evaluate a human’s speed and sudden position changes. For instance, an accelerometer, a gyroscope, or a magnetometer may be used for fall detection [5,6,7]. There are also some commercially available devices of this type [8,9,10]. Although wearable sensors have high accuracy, they are highly prone to false alarms [11]. Additionally, if a person forgets to put on the device, this method becomes useless. Sometimes, wearable devices do not provide automatic fall detection, and the user should trigger an alarm button to call for help in case of falling. However, this becomes impossible if the person gets unconscious after the fall.

Another group of fall detection sensors includes ambient device-based methods based on sensors installed in the room such as acoustic sensors [12], floor pressure sensors [13,14], video cameras [15,16,17], etc. Ambient devices are costly, dependent on their number and placement [4], and may be sensitive to surrounding conditions, such as uniform illuminance, vibration, and the presence of domestic animals, which results in false alarms and affects accuracy [18]. Additionally, in the case of methods based on video surveillance, there are privacy issues, which harden their usage at home (especially in bedrooms and restrooms, where many falls occur) as information leakage may cause unintentional consequences.

Finally, a sensor-fusion method that gets information simultaneously from sensors of different types [19,20,21,22] allows us to overcome disadvantages if using each information channel separately, e.g., to lower the rate of false alarms and to compensate for the dependence of efficiency on the surrounding conditions. However, in the case of the video camera, even if it is used in combination with other sensors, there are still privacy issues. Moreover, additional sensors mean higher costs for the system.

Therefore, to overcome the drawbacks of all the above-listed methods, in this work we have chosen bioradiolocation as an inexpensive, privacy-saving, independent of lighting conditions, and contactless method. Bioradiolocation-based systems have already proven their effectiveness in unobtrusive fall detection [23,24,25]. However, since these systems are currently in the development stage, many aspects are still due to be researched.

One of such challenges in bioradiolocation-based methods is that the fall detection precision strongly depends on the fall direction in relation to the bioradar placement. In order to cover all possible falling directions, some attempts have been made to use an array of bioradars that covers the entire room.

For instance, Maitre et al. [26] used three ultrawideband (UWB) radars for fall detection in realistic conditions. The mean accuracy and Cohen’s kappa for one-radar detection (leave-one-out validation) were 0.87 and 0.76, respectively, whereas accuracy and Cohen’s kappa for the three-radar setup (file-independent validation) were 0.85 and 0.70, respectively. However, for the radar in position 3 in one-radar detection, the mean accuracy and Cohen’s kappa were 0.83 and 0.68, respectively. Additionally, if using data of the closest radar for the classification, Cohen’s kappa values were 0.95, 0.91, and 0.87 for the radars in positions 1, 2, and 3, respectively. This indicates the position dependence of fall recognition using the UWB radar and shows that the multi-radar system may compensate for such a drawback. However, we should consider these results cautiously due to the relatively small sample size (10 volunteers).

Saho et al. [27] addressed the question of fall detection in the restroom by means of the dual-radar system. Doppler radars were mounted onto the ceiling and the wall of the restroom, above and behind the volunteer. Participants (21 men, 22.4±1.1 y.o.) acted out eight types of movement, including the motion of falling forward from a seated position. Using the short-time Fourier transform and the convolutional neural network accuracy values for the ceiling radar, the wall radar and the dual-radar system were 0.90, 0.92, and 0.96, respectively. Therefore, the authors conclude that the best accuracy is obtained for the two radars usage because in this case differences between falls and other types of movement activity can be captured in both upward and horizontal directions simultaneously. Additionally, the sensitivity and specificity of one-vs.-all classifications for the falling movement are 1.00 for either single or dual-radar systems. Therefore, in real-world applications where we would like to detect falls rather than other types of movement activity, fewer radars will allow us to achieve both the appropriate quality classification performance and lower costs.

Yang et al. [28] addressed the question of the radar placement in the case of the single or dual-sensor system. Using raw radar signal and the short-time Fourier transform the micro-Doppler signature was obtained and after feature extraction support vector machine classifier was built. Three radars were used, the first one was mounted on the ceiling at a height of 228 cm with a downward sloping angle of 20°, the second one was installed on the table at a height of 100 cm, and the last one was put on the floor. From the achieved results the authors conclude that placed on the floor the radar signature has the most distinct features. However, in a realistic, complex indoor environment, there will be an occlusion problem. Therefore, in the case of a single sensor, a wide beam radar should be installed vertically down on the ceiling to capture the falling motion in the appropriate direction. However, in the case of two sensors, the better choice is to place the other radar as low as possible avoiding blocking the radar sight.

Mager et al. [29] demonstrated the efficiency of a multi-sensor, radio frequency (RF) system. They used a network of 24 RF nodes installed around the room, whereas the lower level was mounted 17 cm above the floor and the upper level was placed 140 cm above the floor (the upper part of the human torso). Determining the vertical position and motion of a person inside the network, attenuation of radio tomographic images on each layer was obtained and the current position of the person (standing, mid-position, or lying down) was estimated. Then the fall was detected using the time interval between “standing” and “lying down” positions. However, due to the great number of sensors, such a system is quite expensive.

Therefore, the purpose of this work is the evaluation of the optimal configuration of the view-independent bioradiolocation system in order to achieve the best fall detection performance without cost-related problems. We also aimed to compare three- and four-bioradar systems with the previously used two bioradars [30], which did not fully solve the aforementioned problems. The novelty of the present work lies in the proposed multi-channel non-contact and compact bioradar system architecture which allows precise, remote, and view-independent fall detection in human subjects. This approach made it possible to detect any type of fall in any direction without the need to significantly increase the size of the radar system. Thus, the present work contributes to the development of contactless healthcare systems that can be used in combination with smart home technologies.

## 2. Materials and Methods

### 2.1. Experimental Setup

The present work flows organically from our previous research [23]. However, here, we conducted experiments by means of four bioradars whose architecture was proposed in the work [31]. The photo of the bioradar prototype is shown in Figure 1.

The architecture of the bioradar is based on a single-chip high sensitivity quadrature transceiver K-LC5 (RFbeam Microwave GmbH) [32] with two output channels, i.e., the in-phase (I) and quadrature (Q) channels. As far as this radar module does not have an integrated amplifier, it may be used in various applications such as vital sign monitoring, speed and movement detection, etc. Therefore, since we addressed a human fall detection problem, we designed the scheme of the customized amplifier in accordance with the recommendations of the manufacturer [32]. Technical characteristics of the bioradars are shown in Table 1.

Figure 2 presents the diagram of the experimental setup. As it can be seen, bioradars emit continuous electromagnetic wave that interacts with all objects near them. Specifically, reflecting from the moving object, the probing signal (the RF wave) gets phase modulation. Therefore, by processing this signal it is possible to extract information about the object’s displacement. In the case of a biological object, such as a human, the motion of the body surfaces reflects the heartbeat, respiration, and movement activity, including falls.

The reflected RF signal then passes through the amplifier and after that the signal is digitized with the 16-bit analog-to-digital converter (ADC). The Arduino UNO board is used as a microcontroller unit (MCU) that sends the digital signal to the personal computer (PC) for further processing which will be discussed in the next sections in detail. The MCU also controls the bioradar probing frequency which is essential in case of multiple bioradars usage as we would like to avoid interference between the probing signals of the transceivers.

All the bioradars are connected to the same PC, therefore, there is no need for additional synchronization. Additionally, the bioradars are independent of each other and the synergetic effect is significant at the level of classification only, i.e., predictions made with data of each bioradar have equal weights in voting whether this sample is from the positive class or not. In this work, the soft voting technique was chosen for the final decision of the system. This procedure returns the class label as argmax of the sum of predicted probabilities, therefore, it is applicable to both dual-channel systems and multi-channel bioradar systems.

### 2.2. Description of the Experimental Procedure

In the present work, we aimed to evaluate how the variations of view angles and the number of the bioradars affect fall detection performance. Therefore, we used four bioradars and placed them as it is shown in Figure 3. All the experiments were conducted in a standard-sized apartment room (5 × 3 m) with standard furnishing. In accordance with the previous study [30], the bioradars have been mounted 0.5 m above the floor and in such a way that the widest aperture (80${}^{\circ }$) was oriented horizontally to capture as much room space as possible. Bioradars position adjustments, including their relative shift and angular turning, were made in an attempt to minimize an overlapping effect which results in a particularly noisy signal.

Experiments were carried out from January until March 2022 with the participation of eight healthy subjects (four males and four females, 18–24 y.o.). Each of them gave their informed consent prior to the study. For the experiments involving human participants, ethical approval was obtained on 1 March 2018 from the ethics committee of Bauman Moscow State Technical University.

During the experiments, volunteers were asked to act falls from a standing position caused by sleep or loss of balance. Forward, backward, and lateral falls were distributed equally in the sample. The volunteers also performed two other types of high-intensity movements, i.e., squatting or jumping. The experimental scenario was as follows.

A volunteer enters the room and stands still in the center.A recording starts.The volunteer waits for two, four, six, or eight seconds (on his/her own) and then performs one type of high-intensity movement or acts out falling down on the mattress. If the fall type of movement has been chosen, the volunteer keeps lying on the mattress until the end of the recording.The recording ends in 10 s after its start.The volunteer returns to the initial position.

The duration of every bioradar record was 10 s. In total, 400 recordings for each bioradar have been obtained (200 records of falls and 200 records of high-intensity movements). They were labeled as “Fall” if the record contained the falling episode and “NonFall”, otherwise. Volunteers were asked to wait two, four, six, or eight seconds on their own before acting any movement, which allowed us to keep the start time of actions distributed equally along the record duration. This was done to avoid overfitting of the classification models and helped to improve their generalization properties.

### 2.3. Signal Processing Technique

Since 2014, machine learning has become a new mainstream in fall detection algorithms [33]. However, the authors of the review note that classic approaches, such as support vector machines and decision trees are more dominant than deep neural networks due to the limited size of datasets for fall detection. However, using artificial neural networks there is no need for feature extraction and feature selection which may be a cumbersome task. Additionally, the problem of lack of data may be solved to any extent by means of transfer learning [34]. Therefore, as far as in the present work we also have not had enough data for training a large artificial neural network, we have chosen a pre-trained machine learning model, AlexNet [35] specifically, and fine-tuned it using our dataset. Though AlexNet [35] is a relatively old CNN architecture, we have chosen it as it is not very deep. This fact allows for building a shallow model and avoiding unnecessary complications of it.

Then, the next step following data collection is signal processing, which consists of preliminary signal processing, training of machine learning models, and validation of them. Signal processing has been performed with Python 3.7 using PyCharm IDE, while the model’s fine-tuning was implemented in MATLAB R2020b.

#### 2.3.1. Preliminary Signal Processing

The bioradar signal sent from the transceiver to the PC still consists of two separate components, i.e., signals of I and Q channels. Because of the null and optimal detection point problem, there is always one quadrature that ensures the best sensitivity [36]. However, there are various detection schemes, allowing to eliminate the aforementioned problem in systems with quadrature demodulation. Unlike the previous work [23], here, we have used a non-linear arctangent demodulation technique [33] which allows us to extract the phase of the signal. Examples of the signal are shown in Figure 4.

It is worth noting that Figure 4 specifically shows the pairs of signals that have the most distinct differences from each other. For such signal pairs, the use of machine learning methods would be redundant, nevertheless, in the experimental sample, there are also a significant number of pairs with no such clear differences between their patterns (Figure 5). It is for this purpose that this paper uses more in-depth methods than, for example, a fast and simple heuristic algorithm.

#### 2.3.2. Transformation of the Signal

Since AlexNet [35] is a convolutional neural network (CNN) which we have chosen as a base model, the one-dimensional phase of the bioradar has had to be transformed into the image. There are various methods of doing such a transform, however, here, continuous wavelet transform (CWT) has been applied as it allows to get a time-frequency representation (a scalogram) of the signal and to capture its specialties in different frequency bandwidths simultaneously that is useful for processing of non-stationary signals. Additionally, CWT has proved its efficiency in various tasks, such as vital signs monitoring and motion detection by means of continuous wave radiolocation [37,38].

As a mother, wavelet Morlet wavelet with a number of voices per octave equal to 12 was used, as far as it is similar in shape to the bioradar signal. In Figure 6, the scalograms of the phase of the bioradar signal with fall and with other high-intensity movements are shown.

Scalograms were then used as inputs for the CNN, which was used as a classifier to perform fall detection. Therefore, by means of CWT, we proceeded from the time series analysis to the image classification problem.

#### 2.3.3. Training of Machine Learning Models

Previously obtained scalograms were resized to fit the input size of AlexNet [35] (227 × 227 × 3 pixels) and additionally augmented using a left-right shift of 50 pixels to prevent the CNN from overfitting and memorizing particular details of the images.

The next step is the CNN modification and fine-tuning. The original AlexNet [35] was trained to classify images for 1000 classes, though, in this work, there is a binary classification problem (Fall/NonFall). Therefore, we replaced the CNN’s last fully connected layer with another one with an output size of two and randomly initialized weights which were fine-tuned further in the training process. The final architecture of the CNN is shown in Figure 7.

In the present work, the performance of the models was evaluated using leave-one-out and hold-out validation which will be discussed in detail in the next section. Since we did not adjust any of the CNN’s parameters, there was no validation dataset.

## 3. Results

First of all, to achieve the goals of this study, eight AlexNet [35] CNN models were trained to estimate the volunteer-to-volunteer variation of fall detection performance. The performance of the models was evaluated using leave-one-out validation with volunteers’ separation, i.e., data of seven distinct participants were used for model fine-tuning, whereas the record of the remaining participant was used as a test dataset. We did not adjust any CNN parameters, therefore, there was no validation dataset. From the data of all four bioradars, each training and test dataset contained 1400 and 200 images, respectively. Then, a confusion matrix for the models was constructed and four well-known metrics for the numerical evaluation of performance were calculated, such as accuracy, sensitivity, F1-score, and Cohen’s kappa (Figure 8).

The obtained results show a not-so-substantial variety between the volunteers. Slightly worse results among the last three participants were due to a significant false-negative rate, which could be explained by the volunteers’ lack of experience in conducting similar experiments, so some of the fall activities were of a lower intensity and were more similar to the high-intense “NonFall” movements.

In the next step, the sample with the best overall performance (Sample No.4) was used for the evaluation of the performance of the 2-, 3-, and 4-bioradar systems of different configurations.

The decision to classify each entry follows the algorithm described below.

At the input of each of the four trained models the data from the corresponding radar is fed.Each model simultaneously and independently classifies “its” data.The final output class of the image is determined by the majority of votes. In case the votes are equally divided, the total weight of the Softmax layer is calculated, and the output class is determined by the maximum obtained sum.

All in all, eight additional models for the various bioradar system configurations have been constructed, they are

four models for the pairs of No.1&No.2, No.1&No.4, No.2&No.3, and No.3&No.4 bioradars, andfour models for the trios of No.1&No.2&No.3, No.1&No.2&No.4, No.1&No.3&No.4, and No.2&No.3&No.4 bioradars.

Then, similarly to above, a confusion matrix for each model was constructed and four statistical metrics for the numerical evaluation were calculated. Each system, in general, has shown decent classification performance in terms of the average scores (Table 2). However, in this study, it is more important to observe the minimum scores achieved for a particular subject during system evaluation, since this particular data would allow us to evaluate the consistency of fall detection performance. Thus, the minimum achieved values for a 4-bioradar system are shown in Table 3, whereas minimum values for 2- and 3- bioradar systems are graphically shown (due to the increased amount of data) in Figure 9 and Figure 10.

The achieved data illustrates the key drawback of using two- and three-bioradar systems. Despite the fact of being able to show performance on par with the 4-radar system in certain cases, those systems cannot perform consistently well with the detection of all kinds of falls. This may be not only due to the relatively poor ability to detect falls in certain directions (e.g., diagonally away from the pair of radars) but also to the smaller training dataset in comparison with a 4-bioradar system.

To visualize the differences between 2-, 3-, and 4-radar systems, the overall average of minimal performance values of each model has been calculated (Figure 11). It can be seen that the use of four bioradars allows for achieving considerably better results in the fall detection task. Meanwhile, comparing the improvement of performance between 2-bioradar and 3-bioradar systems versus between 3-bioradar and 4-bioradar ones, the former is more significant than the latter. Therefore, there is a trade-off between performance and costs caused by the number of channels. In future work, the question of an optimal number of channels in terms of the highest performance and the lowest cost should be addressed.

## 4. Discussion

A comparison of our results with other works is shown in Table 4. We should note that due to the specifics of the task there is no universal benchmark dataset. Therefore, a direct comparison of the works would not be correct. Moreover, most radar datasets are relatively small consequently we should interpret the results cautiously.

As we see, our results are among those that are the state of the art. A continuous wavelet transform allowed us to extract a valuable representation of the classes while AlexNet CNN and the implementation of transfer learning lead to the rapid construction of lightweight models. This will enable real-time use of the system and the ability to quickly fine-tune the model using custom data from where it will actually be used. Moreover, the proposed method has advantages such as privacy protection, invariance to lighting variations, and opaque obstacles. Additionally, there is no need to wear any sensors.

However, further improvement of the fall detection performance by using five or more radars may be controversial. It may well be that the use of four radars is already sufficient for the achievement of the view-independent approach in the task of fall recognition since such a system allows one to achieve a field of view very close to the 360°. Furthermore, as shown in the results section of this article, there is a trade-off between the accuracy of the system and total cost, which increases with the number of sensors.

Therefore, in realistic conditions, the possible way to determine an optimal configuration of the system is to solve the problem of the height of sensors and their angle orientation in relation to the floor.

It should be noted that in this work the dataset was artificially balanced. Although the data used for model training in fall detection tasks by means of radiolocation typically contain as many falls as there are episodes without falls [23,26,27], it should be borne in mind that this is an unnatural situation in real life when falls are rare events (once a month, year, etc.). Therefore, even small changes in specificity may lead to a large number of false positives in real applications. This drawback can be eliminated by including an additional information channel in the system, e.g., a camera, which will increase the specificity of the system as a whole (fusion technique). Another possible approach is voice feedback from the user which can function as follows.

The system detects a fall and sends a voice message to the user, such as “Did you fall? Do you need help?”In case the user replies that he/she is fine, the alarm is turned off and the system continues to work normally.In case there is no response within 60 s, or the user says that he needs help, the system will send an alarm message to the user’s relatives or an ambulance.

However, in terms of long-term consequences for the user, it is better to detect a false fall than to miss a true one.

Finally, there is still a lack of investigation of the system performance in case there is more than one person in the room or there are moving pets. Thus, all the aforementioned questions should be considered in future work.

## Figures and Tables

**Figure 1 sensors-22-06285-f001:**
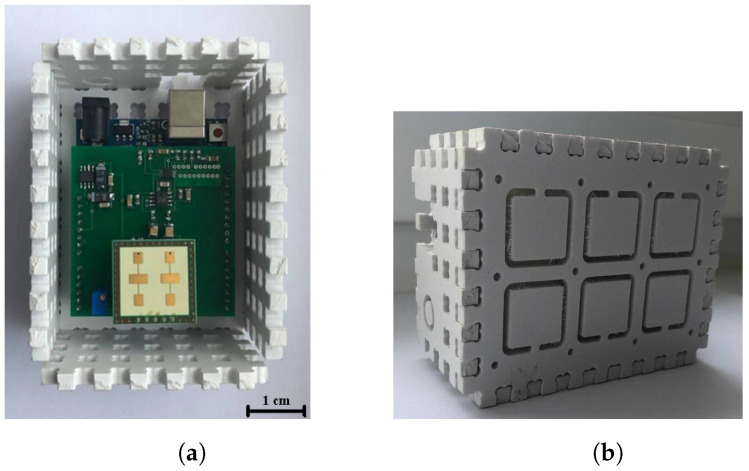
The bioradar prototype photos. (**a**) The front panel removed. (**b**) The bioradar assembly.

**Figure 2 sensors-22-06285-f002:**
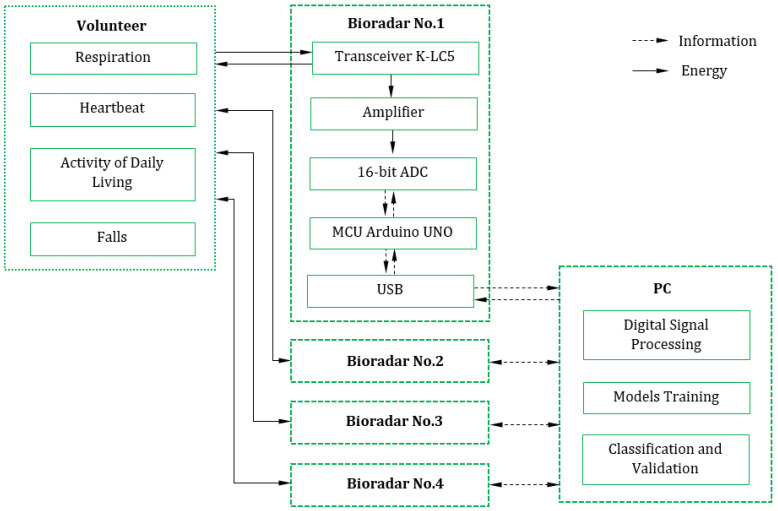
The diagram of the interface between the bioradiolocation system and the volunteer.

**Figure 3 sensors-22-06285-f003:**
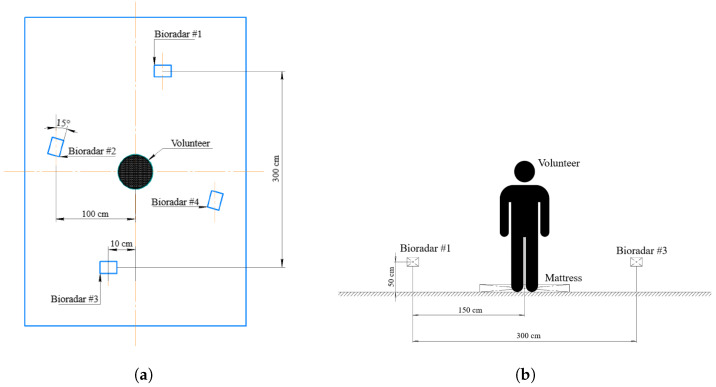
The scheme of the experiment. (**a**) Top view. (**b**) Front view.

**Figure 4 sensors-22-06285-f004:**
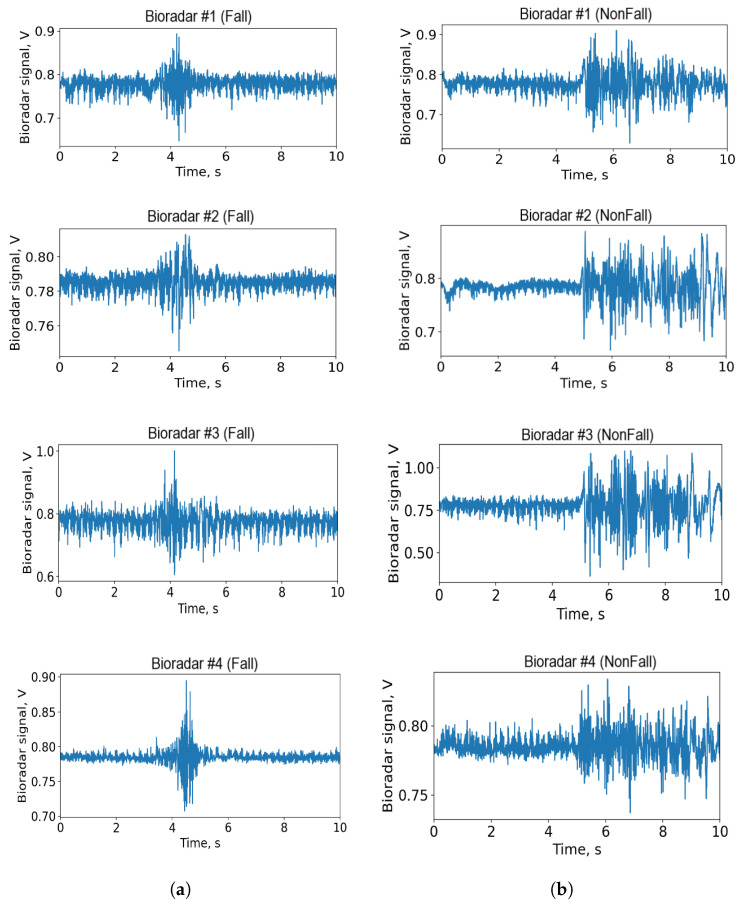
Examples of the preprocessed signal. (**a**) The fall event. (**b**) The squatting episode.

**Figure 5 sensors-22-06285-f005:**
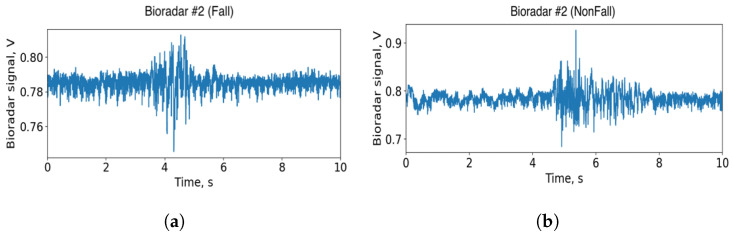
Examples of the preprocessed signal of similar patterns. (**a**) The fall event. (**b**) The squatting episode.

**Figure 6 sensors-22-06285-f006:**
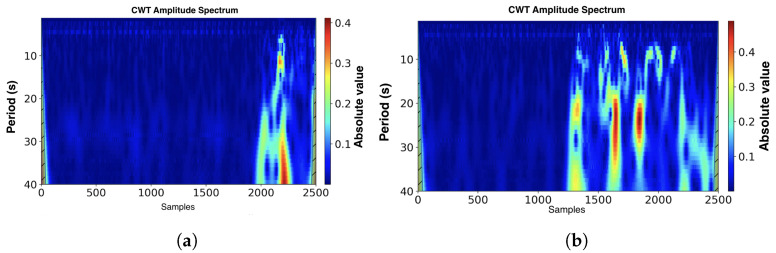
Scalograms of the bioradar signal. (The unit marks were added to the figure. This comment may be deleted.) (**a**) The fall event. (**b**) The squatting episode.

**Figure 7 sensors-22-06285-f007:**
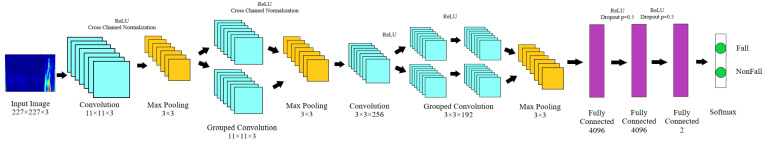
The CNN’s architecture.

**Figure 8 sensors-22-06285-f008:**
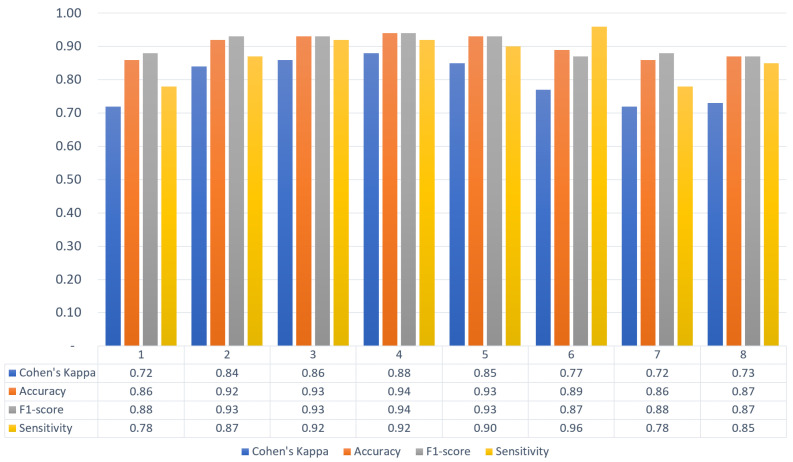
Leave-one-out validation of the models.

**Figure 9 sensors-22-06285-f009:**
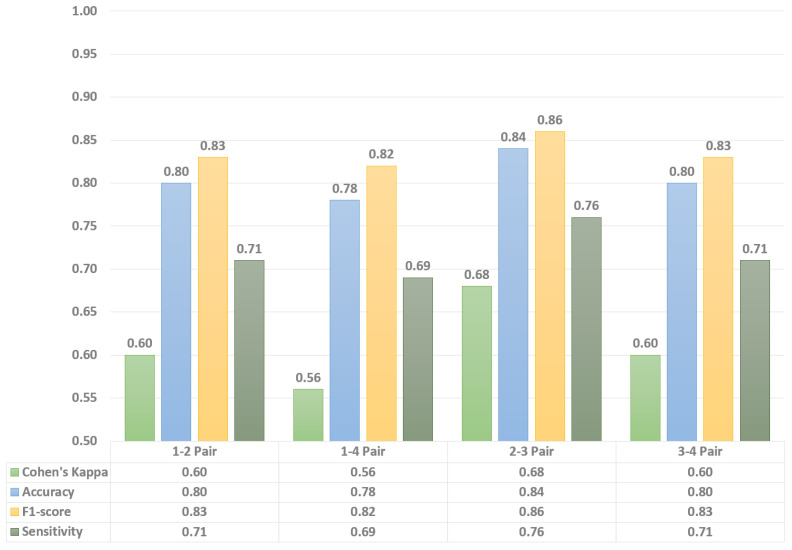
Comparison of performance in 2-bioradar system configurations.

**Figure 10 sensors-22-06285-f010:**
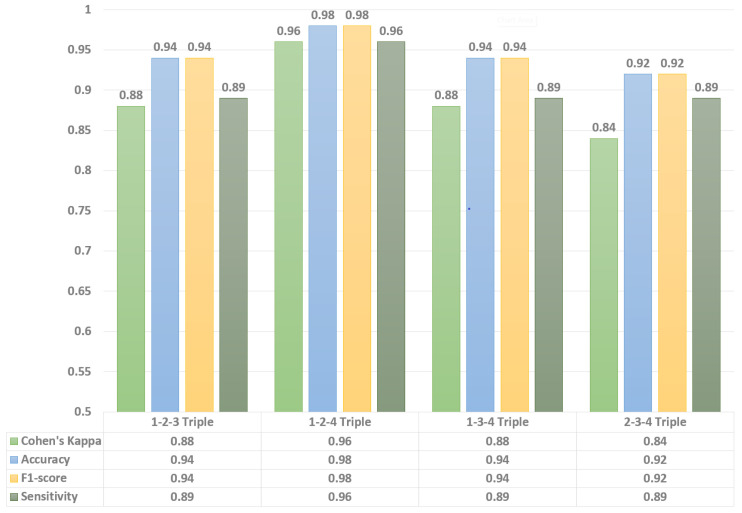
Comparison of performance in 3-bioradar system configurations.

**Figure 11 sensors-22-06285-f011:**
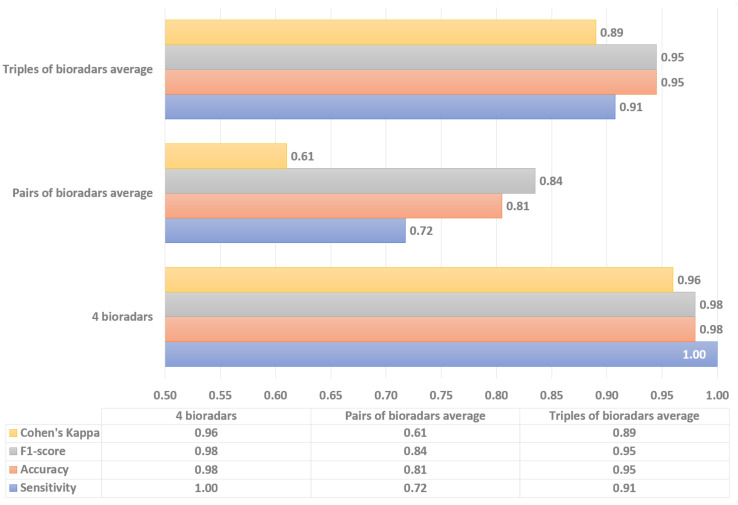
Overall comparison of the system configurations.

**Table 1 sensors-22-06285-t001:** Technical characteristics of the bioradar system.

Parameter	Value
Probing frequency	24.0–24.2 GHz
Type of probing signal	Continuous wave
Detecting signal band	1–100 Hz
Gain	15–30 dB
Radar range	0.5–6.0 m
Radiated power density	<3 μW/cm${}^{2}$
Antenna’s beam aperture	80${}^{\circ }$/34${}^{\circ }$
Size (for a single radar block)	95 × 75 × 45 mm
Number of channels	4

**Table 2 sensors-22-06285-t002:** Average classification scores.

Number of Bioradars	Sensitivity	Accuracy	F1-Score	Cohen’s Kappa
2	0.85	0.92	0.93	0.86
3	0.97	0.98	0.98	0.97
4	1.00	0.99	0.99	0.99

**Table 3 sensors-22-06285-t003:** Minimum achieved scores for the 4-bioradar system.

Sensitivity	Accuracy	F1-Score	Cohen’s Kappa
1	0.98	0.98	0.96

**Table 4 sensors-22-06285-t004:** Comparison between the achieved results and other works.

Ref.	Sensor Type	Number of Channels	Techniques	Number of Subjects	Accuracy (%)
Lin [7], 2020	Accelerometer, Gyroscope, Magnetometer	3	Threshold-based algorithm	15	97.8
Fatima [15], 2021	Camera	1	AlphaPose, Recurrent-Convolutional Neural Network	9	77.5
Ramirez [16], 2021	Camera	1	AlphaPose, Random Forest	17	99.3
Anishchenko [23], 2019	Radar	2	CWT and CNN	5	99.3
Maitre [26], 2021	UWB radar	3	CNN-LSTM	10	87.0
Saho [27], 2022	Doppler radar	2	Fourier Transform and CNN	21	100
This work	Radar	2	CWT and CNN	8	92.0
		3			98.0
		4			99.0

## Data Availability

The data presented in this study are available on request from the corresponding author. The data are not publicly available due to privacy issues.

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
