# Peer review of "Contactless Fall Detection by Means of Multiple Bioradars and Transfer Learning"

_sensors, 2022, doi:10.3390/s22166285_

Round 1

Reviewer 1 Report

Dear authors,

minor remarks:

-there are some minor English typos that can be easily corrected.

-it might make sense to place a "reference element" (coin or ruler) next to the figures to give the reader not familiar with radar am impression of the size of the modules

 major remarks:

-please review Fig. 3. Are the units really mm? I guess cm?! This must be corrected I guess

-I'm in generally in doubt about the successfulness of these contactless fall detection methods. I think all in all the number of false alarms will always be too high for practical application, regardless of the numbers of radars etc. In fact, and in reality, you have a highly unbalanced data set. The event of a fall will happen e.g. once a week/month, and you have thousends of measurements where no fall will happen. Thus, even extremely low false alarm rates will cause too much false alarms in practice.  A discussion of this issue would significantly enhance the quality of the paper. Furthermore, this fact should be simulated in the experiment design (much more non-fall events should be simulated and evaluated) However, I know, this issue is neglected in almost all publications.

- when looking at figure 4, it seems to be easy to write an heuristic algorithm (e.g. estimation of the variance of the signal over a window) to detect fall vs. non-fall. At least  some remarks here would be in order. Furthermore, this reflects the problem with experiment design for such application.

As a final remark, and as you correctly wrote, there are lots of open issues (pets, ...) that might prevent the practial application of these techniques in practice.

Reviewer 2 Report

This manuscript investigates a fall detection method based on multiple radars. The simulation results show that this work can achieve fall detection with high accuracy. However, the authors just increase the number of radars and this research is lack innovation. The specific suggestions are as follows.

1.     The contributions of this manuscript should be clarified.

2.     More discussion about the experimental results should be given.

Round 2

Reviewer 1 Report

Your corrections significantly improved the quality of the presentation.

Author Response

We thank the Reviewer for the valuable comments.

Reviewer 2 Report

This manuscript has been revised according to previous comments. However, there are still some questions as follows.

1.     The innovation of this paper should be clarified. There are many fall detection methods based on multiple radars. Please explain why the proposed method is superior to others.

2.     In Figure. 6, the label of the axis and its unit should be added.
